# Endosome Traffic Modulates Pro-Inflammatory Signal Transduction in CD4^+^ T Cells—Implications for the Pathogenesis of Systemic Lupus Erythematosus

**DOI:** 10.3390/ijms241310749

**Published:** 2023-06-28

**Authors:** Joy S. Park, Andras Perl

**Affiliations:** 1Department of Medicine, Norton College of Medicine, State University of New York, Upstate Medical University, Syracuse, NY 13210, USA; 2Department of Biochemistry and Molecular Biology, Norton College of Medicine, State University of New York, Upstate Medical University, Syracuse, NY 13210, USA; 3Department of Microbiology and Immunology, Norton College of Medicine, State University of New York, Upstate Medical University, Syracuse, NY 13210, USA

**Keywords:** CD4^+^ T cells, endosome traffic, lysosome, metabolism, glucose, tryptophan, kynurenine, glutamine, mTOR, interferon, JAK/STAT, IL-2, IL-17, autoimmunity, systemic lupus erythematosus

## Abstract

Endocytic recycling regulates the cell surface receptor composition of the plasma membrane. The surface expression levels of the T cell receptor (TCR), in concert with signal transducing co-receptors, regulate T cell responses, such as proliferation, differentiation, and cytokine production. Altered TCR expression contributes to pro-inflammatory skewing, which is a hallmark of autoimmune diseases, such as systemic lupus erythematosus (SLE), defined by a reduced function of regulatory T cells (Tregs) and the expansion of CD4^+^ helper T (Th) cells. The ensuing secretion of inflammatory cytokines, such as interferon-γ and interleukin (IL)-4, IL-17, IL-21, and IL-23, trigger autoantibody production and tissue infiltration by cells of the adaptive and innate immune system that induce organ damage. Endocytic recycling influences immunological synapse formation by CD4^+^ T lymphocytes, signal transduction from crosslinked surface receptors through recruitment of adaptor molecules, intracellular traffic of organelles, and the generation of metabolites to support growth, cytokine production, and epigenetic control of DNA replication and gene expression in the cell nucleus. This review will delineate checkpoints of endosome traffic that can be targeted for therapeutic interventions in autoimmune and other disease conditions.

## 1. Introduction

The Rab GTPase subfamily, the largest subfamily of the Ras family of GTPases, is composed of more than 60 family members in humans, and they control different steps of receptor trafficking [1]. Alterations in Rab proteins are connected to a multitude of diseases, such as cancer [2,3,4,5], neurodegenerative diseases [6,7,8], and immune disorders [9]. Receptor recycling and endocytic trafficking control the surface expression levels of T cell receptors (TCR) [10,11]. Altered TCR expression levels lead to dysregulated TCR signaling, contributing to proinflammatory skewing, a critical agent of abnormal T cell activation in autoimmune diseases, including rheumatoid arthritis and systemic lupus erythematosus (SLE) [12].

In contrast to pro-inflammatory cytokines, interleukin 2 (IL-2) plays a unique role as an anti-inflammatory cytokine given its fundamental role in regulatory T cell (Treg) differentiation [13]. SLE patients show reduced numbers and impaired function of Tregs [14,15]. Tregs are important CD4^+^ T cells that are identified by the expression of the transcription factor, FoxP3, and the inhibition of autoreactive T cells [16]. Tregs function via direct cell-to-cell contact or by the production of immunosuppressive cytokines, such as transforming growth factor β (TGFβ) and IL-10 [17]. TGFβ and IL-6 are required for the development of helper T 17 (Th17) cells, and IL-6 is increased in SLE [18]. Th17 cells are identified by the specific transcription factor, retinoic-acid-receptor-related orphan nuclear receptor γ (RORγt), and the production of IL-17 [19,20]. The IL-17 signature plays an important role in effector-cell-mediated tissue damage by recruiting other pro-inflammatory cells [21]. Follicular helper T (Tfh) cells are CD4^+^ T cells that maintain the germinal centers that derive autoreactive antibodies secreting plasma cells through the secretion of IL-4 and IL-21 [22]. Tfh cells are abnormally expanded in SLE patients and lupus-prone mice [23]. 

The expansion of Th17 cells and the increased production of IL-17A correlate with disease activity in SLE patients [24]. Similarly, decreased IL-2 production promotes the imbalance of Th17/Treg, which contributes to organ inflammation and damage in SLE patients [25]. Understanding the underlying defects in SLE patients’ CD4^+^ T cells that lead to proinflammatory skewing is important in understanding the disease’s pathophysiology. The following sections describe the disturbed balance of proinflammatory and anti-inflammatory cells in SLE and how they are regulated by their altered receptor endosomal recycling, resulting metabolic abnormalities and downstream signaling.

## 2. Endosomal Trafficking and Recycling Pathways

Cell surface expression of membrane proteins can be modified through changes in vesicular transport that arise from changes in trafficking or the behavior of entire organelles [26]. In response to signaling mechanisms, membrane trafficking changes to increase or decrease the surface expression of proteins [26]. Surface receptor expression can be decreased when stimulated by their ligands, while, for example, in response to insulin, the glucose transporter surface expression can increase [26].

Transmembrane proteins are endocytosed, a process where cargoes (receptors and their bound ligands) are transported in membrane-bound vesicles from the cell surface to inside the cell. There are two main endocytosis pathways, clathrin-independent endocytosis (CIE) and the more well-characterized clathrin-mediated endocytosis (CME) [26,27]. Clathrin is a protein that plays a major role in the formation of clathrin-coated vesicles, which make up to 95% of endocytic vesicles [28]. Once internalized into the cell, regardless of the endocytic pathway, cargoes merge into the common endosomal network: an interconnected “highway” system that controls the trafficking and transfer of cargoes between organelles. The endosomal network collects, sorts, and sends cargoes to their final destinations in the cell [29].

Rab GTPases are the “master regulators” of intracellular trafficking. Rab proteins are localized in distinct compartments, where they recruit effector proteins to regulate the transport between organelles [1,30]. Rab GTPases function as “molecular switches” and cycle between the inactive GDP-bound and active GTP-bound states. Guanine nucleotide exchange factors (GEFs) catalyze the dissociation of GDP from a GTPase for GTP to replace GDP, and GTPase-activating proteins (GAPs) catalyze the hydrolysis of the third phosphate of GTP to create GDP [31]. Since cells internalize their receptors about one to five times an hour [32], regulating Rab GTPase activity is important in retaining vesicle trafficking. 

Internalized cargoes converge into a common early endosome, where they are sorted for subsequent transport to different parts of the cell. The acidic environment (pH~6.5) of the early endosome causes ligands to be released from the receptors. Most of these ligands are sorted into the late endosomes and then into the lysosomes for degradation [33]. The receptors themselves can be degraded in the lysosome, but can also have different fates, such as being transported to the *trans*-Golgi network (TGN) or the plasma membrane for reuse, also known as recycling [34]. While the mechanisms that mediate the vesicular transport along the lysosomal degradative pathway have been well characterized, the regulation of sorting and recycling cargoes is not fully understood.

The Rab proteins are localized to specific endosomes but are segregated to distinct regions of the membrane, defining specialized functional membrane domains (Figure 1) [35]. There are two pathways for recycling back to the plasma membrane: (1) the rapid-recycling pathway, where recycling occurs directly from the early endosome, and (2) the slow-recycling pathway, where recycling occurs indirectly via the endosomal recycling compartment (ERC), a subpopulation of recycling endosomes [35,36]. In the rapid-recycling pathway, receptors are internalized and delivered to the early endosomes that are marked by Rab5, then recycled back onto the cell surface via the recycling pathway under the control of Rab4. Alternatively, during the slow-recycling pathway, cargo proteins are transported from the early endosome to the ERC, then back onto the cell membrane. Rab11 is localized to the ERC and TGN and is important in the slow-recycling pathway [35]. In other parts of intracellular trafficking, Rab9 is localized to the late endosome and TGN [37]. Rab7 is found in late endosomes and lysosomes, and Rab5 and Rab7 are transition cargoes from early endosomes to late endosomes [38]. 

The composition of the plasma membrane is controlled by the balance between endocytic uptake and recycling, contributing to physiological functions such as nutrient uptake and signal transduction [27,34]. Cell surface proteins that are recycled include proteins involved in nutrient uptake, such as the transferrin receptor (CD71) (for iron), glucose transporters (GLUT), and lipoprotein receptors (for cholesterol); cell adhesion molecules (integrins and cadherins); and cell signaling proteins (i.e., ErbB family proteins) and G protein-coupled receptors (i.e., chemokine receptors) [26,33,39,40,41,42].

Rab5 is required for the formation and function of the early endosome [43,44]. Every transport step requires the GTP-bound, activated Rab GTPases to bind to effector proteins [1]. For Rab5, upon activation by its effector proteins, the Rabaptin-5/Rabex-5 complex [45], it recruits phosphoinositol-3 kinases (PI3K) [46], including hVPS34 [47], subsequently generating phosphoinositol-3-phosphate (PI3P). An environment of PI3P and Rab5 is needed for the docking protein, early endosomal antigen 1 (EEA1), to bind to the endosomal membrane [48,49,50,51]. Rab5, together with EEA1, regulates the fusion between primary endocytic vesicles and sorting endosomes [52,53,54]. Rab35 colocalizes with CD71 on the plasma membrane and vesicles, regulating the rapid recycling of CD71 [55].

Rab4, localized to the early endosomes, recycles glycosphingolipids from early endosomes through rapid recycling [56], and its dominant-negative form inhibits rapid recycling [57]. Rab4 transfers cargoes into either recycling or degradative pathways [58,59]. The HTLV-1 related endogenous retroviral sequence (HRES-1)/Rab4, also designated as Rab4A, is overexpressed in SLE patients’ T cells [10], and it regulates the expression of CD71 [60,61], CD4 [61], and TCRζ [10]. When Rab4A is overexpressed in T cells, endocytic recycling of TCRζ and CD4 is inhibited and targeted for lysosomal degradation [10,61]. The loss of TCRζ and CD4 in SLE T cells changes their downstream TCR signaling pathways [10]. The composite description of endocytic recycling may vary between immune cell types and may affect their functions through downstream pathways through changes in metabolites.

### 2.1. Endocytic Regulation of Antigen Presentation to CD4^+^ T Cells

CD4^+^ T cells are activated through the engagement of the TCR with antigens presented by major histocompatibility complex (MHC)-II molecules [62]. The TCR’s affinity to the self or the agonist peptide-MHC II complex is important in the fate of CD4^+^ T cells in that it determines whether CD4^+^ T cells are differentiated into naïve CD4^+^ T cells or Tregs [63]. Recent studies addressed the role of Rab5, Rab7, Rab9, and Rab11 GTPases in MHC-II-dependent antigen presentation by dendritic cells [64]. MHC-II–peptide complexes are associated with each of these Rab GTPases during dendritic cell maturation, while MHC-II complexes sequentially traffic from Rab5^+^, Rab7^+^, and Rab9^+^ early endosomes to the cell surface via Rab11^+^ endosomes [64]. MHC-II can also be secreted from dendritic cells in exosomes [65]. Both Rab4A [66] and Rab4B are involved in MHC-II-mediated antigen presentation by B cells [67].

Exosomes are extracellular vesicles that are released from the MVB when it fuses with the plasma membrane [68]. Endosomal sorting complexes required for transport (ESCRT) proteins [69], and Rab27a and Rab27b GTPases play an important role in their biogenesis [70]. Exosomes can contain nucleic acids, lipids, metabolites, cytosolic proteins, and cell surface molecules [71]. Exosomes from antigen presenting cells (APCs) play an important role in presenting extracellular antigen to CD4^+^ T cells to promote T cell immunity during infection [72]. T- and B cell-derived exosomes have been found to be involved in SLE, for example, miR-451a expression in CD4^+^ T cell- and B cell-derived exosomes is downregulated in SLE patients with active disease and correlates with renal damage [73].

### 2.2. Contribution of Endosomal Traffic to T Cell Synapse Formation

When a T cell recognizes an antigen presented by an APC, the TCR on the T cell organizes an immune synapse, involving the TCR/CD3 complex and co-receptor CD4 [74]. The TCR is a heterodimer, commonly consisting of the α and β chains, which recognizes antigens presented by APCs on their MHC-I or -II [75]. The TCR is assembled with a complex of CD3 proteins with the subunits δ, ε, γ, and ζ [76]. The co-receptor, CD4, also binds to MHC-II [77].

When the TCR engages with a peptide–MHC II complex presented by an APC, signaling molecules are recruited via endocytic traffic to the site of interaction, which is termed the immunological synapse (IS). The IS is also called the supramolecular activation complex (SMAC), which is comprised of concentric structures, designated as central, peripheral, and distal, or c-SMAC, p-SMAC, and d-SMAC, respectively [78]. The c-SMAC contains the TCR, CD3, CD4, and CD28; the p-SMAC contains adhesion molecules, lymphocyte function-associated antigen 1 (LFA-1) and intracellular adhesion molecule (ICAM) molecules, and signal transducers such as CD71 [79]. The movement of CD4 and CD71 in and out of the IS is triggered by the activation of protein kinase C (PKC) [80] and mediated via endosome traffic by Rab4A [61]. Signal transduction from the CD3ζ chain depends on the phosphorylation of tyrosine residues by the lymphocyte-specific protein tyrosine kinase (Lck), which is recycled via endocytic traffic controlled by Rab11 and Rab11 family interacting protein-3 [81]. CD3ζ has three immunoreceptor tyrosine-based activation motif (ITAM) domains. Phosphorylated ITAM domains recruit ζ-associated protein kinase 70 (ZAP-70), which becomes phosphorylated and activated by Lck [82]. Activated ZAP-70 phosphorylates the adaptor proteins, LAT, and lymphocyte cytosolic protein 2 (LCP2 or SLP-76), which then bind to and activate phospholipase cCγ (PLC-γ). Activated PLC-γ hydrolyzes phosphatidylinositol-4,5-biphosphate (PIP2), producing inositol 1,4,5-triphosphate and diacylglycerol. This results in calcium flux, and PKC becomes activated. Then, Ras guanine releasing protein 1 (RasGRP1) is recruited, and the Ras-mitogen-activated protein kinase pathway is activated [83,84] (Figure 2).

Moreover, endosome traffic delivers other essential cargoes for the formation of a functional IS on CD4^+^ T cells, such as the linker for the activation of T cells (LAT1), the guanine exchange factor Vav, and several actin polymerization regulatory proteins, like the Wiscott–Aldrich syndrome protein (WASp), the WASp-interacting protein, WAVE-2, and coronin-1 [85]. Following activation, CD3, CD4, and CD71 are internalized by Rab5^+^ endosomes to be redirected to the cell surface in Rab4^+^ or Rab11^+^ early endosomes [86]. When the TCR engages self-antigens, death receptors, such as CD95/Fas/Apo1, become activated and trigger a wave of endosomal traffic to the mitochondria, thus initiating apoptosis [87]. The retrograde endosomal traffic of LAT1 between the IS and the TGN is controlled by Rab6 and Syntaxin-16 [88]. In turn, the association with sorting nexin 27 (SNX27) promotes endosome rerouting to the IS by preventing its traffic to lysosomes [89]. Notably, SNX27^+^ sorting endosomes are associated with Rab4 [90]. Future studies may be aimed at the role of SNX27 in directing Rab4-mediated endosome traffic between the cell surface and lysosomes.

## 3. Trafficked Receptors Impact Metabolic Abnormalities in T Cells

In autoimmune diseases, such as SLE, changes in the expression of several surface proteins are linked to the functional defects in T cells [91]. Surface receptors on T cells promote signaling cascades that determine the cell’s fate, cytokine production, and differentiation. When T cells differentiate and execute effector functions, they undergo metabolic reprogramming [92]. In SLE, T cell metabolism is dysregulated in multiple ways [93]. Therefore, it is important to link how the expression levels of surface proteins that are regulated by endosomal recycling may lead to metabolic abnormalities found in SLE.

### 3.1. TCR and STIM1 Trafficking Affects Calcium Flux

T cell activation causes CIE of the TCR/CD3ζ subunit (TCRζ), which has decreased expression in SLE T cells [94]. Previous work in our lab has shown that the Rab4A-dependent lysosomal degradation contributes to the loss of TCRζ and CD4 in SLE T cells [10] (Figure 1).

T cells from SLE patients are marked by aberrant TCR signaling, causing hyperresponsiveness of T cells [95]. Changes in the expression of TCRζ caused by the receptor trafficking and recycling pathways have been found to be behind this phenomenon [10]. In place of the CD3ζ protein, Fc epsilon receptor I gamma chain (FcεRIγ) is substituted, which is not normally expressed in resting T cells [96]. CD3ζ and FcεRIγ are structurally homologous [97]. In SLE T cells, FcεRIγ recruits Syk instead of ZAP-70. The interaction between FcεRIγ and Syk is significantly stronger than the normal interaction between CD3ζ and ZAP-70, increasing calcium influx in these T cells [98,99,100] (Figure 2). Interestingly, increased intracellular calcium concentration in T cells can initiate exosome secretion [101]. In SLE T cells, activation accompanied by calcium influx play a role in cytokine production and inflammatory lineage development through multiple pathways.

Furthermore, TCR-induced calcium fluxing involves the trafficking and activation of calcium release-activated calcium channels (CRACs) [102]. CRAC1 is a calcium-selective ion channel protein that is encoded by the ORAI1 gene [103]. The inactivation of CRAC1 due to genetic mutations causes severe combined immunodeficiency (SCID). Stromal interaction molecule 1 (STIM1) is a transmembrane protein that is mainly localized to the endoplasmic reticulum, where it senses calcium depletion, which triggers its translocation to the plasma membrane and association with ORAI1 [104]. Of note, Rab4 and Rab5 regulate the endocytic traffic of STIM1 and thus modulate calcium flux via ORAI1 [105]. Calcium current through CRAC can be inhibited pharmacologically in CD4^+^ T cells by RO2959 [106]. Similar to STIM1, another calcium binding protein, CRAC regulator 2A (CRACR2A) also possesses an EF-hand, as well as Rab GTPase activity, that acts as an adaptor for dynein and transports intracellular cargoes [107]. CRACR2A is a cytosolic Ca^2+^ sensor that traffics STIM1 to the plasma membrane to ORAI1. Biallelic CRACR2A mutations also cause immune deficiency with autoimmune inflammatory complications [108].

### 3.2. Glucose Transporters and Metabolism 

There are two steps to glucose metabolism: (1) the breakdown of glucose and conversion into pyruvate and (2) the TCA cycle to fuel OXPHOS in the mitochondria [109] (Figure 2). The primary metabolic pathway for Th17 cells is aerobic glycolysis, while for Tregs, it is OXPHOS through fatty acid oxidation [110,111]. 

Glucose transporters (GLUT) mediate glucose uptake, and for T cells, GLUT1 is the major glucose transporter [112]. IL-7 promotes glycolysis and GLUT1 trafficking [113]. In ovarian cancer cells, Rab25 regulates the recycling of GLUT1 to the cell surface [114]. Th17 cells have higher GLUT1 surface expression than Tregs [115,116]. In SLE, increased GLUT1 expression in CD4^+^ T cells is associated with increased activation and IL-17 production [117]. In mouse models, depleting GLUT1 reduces Th17 differentiation, while Treg differentiation remains unchanged, improving inflammatory bowel disease and GVHD [115]. Similarly, treating lupus-prone mice with CG-5, a glucose transporter inhibitor, reduced Th17 differentiation while promoting Treg differentiation, improving disease activity [118]. The glucose-dependent metabolism activates the calcium signaling pathway through generating phosphoenolpyruvate (PEP) during glycolysis [119] (Figure 2). 

Glucose metabolism provides two essential metabolites for cell growth and survival via the pentose phosphate pathway (PPP): (1) ribose 5-phosphate for the synthesis of nucleotides and (2) NADPH for the synthesis of lipids and the maintenance of a reducing environment. NADPH can directly or indirectly neutralize reactive oxygen species (ROS) via regenerating reduced glutathione (GSH) from its oxidized form, glutathione disulfide (GSSG) [120]. The PPP enzyme transaldolase regulates the availability of NADPH and intracellular GSH in CD4^+^ T cells [121]. NADPH and GSH levels impact signal transduction via the control of the mitochondrial transmembrane potential (∆Ψm), an early checkpoint of T cell activation [122]. Intracellular GSH is depleted in lupus T cells, resulting in mitochondrial hyperpolarization (MHP) and adenosine triphosphate (ATP), which predisposes to pro-inflammatory cell death via necrosis [123,124,125]. MHP is linked to mTOR activation, which can be reversed by replenishing GSH by providing its rate-limited antioxidant precursor, N-acetylcysteine (NAC) [126]. Increased ROS production and GSH depletion has been attributed to the accumulation of mitochondria due to deficient recycling via mitophagy [127]. In turn, deficient mitophagy is caused by the Rab4A-mediated depletion of mitochondrial fission initiator dynamin-related protein 1 (Drp1) [127]. The targeting of Drp1 to lysosomal degradation is reversible by inhibiting the enzymatic activity of geranylgeranyl transferase II with 2-hydroxy-2-phosphono-3-(pyridin-3-yl) propanoic acid (3-PEHPC), which also provides preliminary evidence for therapeutic blockade of glomerulonephritis in lupus-prone MLR/lpr mice [127].

### 3.3. Transferrin Receptor, CD71, and Iron Metabolism

Iron plays an important role in hemopoiesis and mitochondrial function [128]. Excess iron can impair mitochondria, and it is one of the metabolic abnormalities found in SLE T cells [129]. Two iron molecules bind to transferrin, a soluble transporter protein. Once iron binds to transferrin, it binds to the transferrin receptor, CD71, on the cell surface [130]. Then, CD71 becomes internalized and fuses with the endosome. Ferrous ion leaves the endosome via an iron transporter, divalent metal transporter 1 (SLC11A2, DMT-1), into the cytoplasm [129] (Figure 2).

SLE patients’ CD4^+^ T cells contain higher intracellular iron levels than those of healthy controls [131,132]. Increased iron levels are correlated with epigenetic changes that promote Tfh differentiation [132]. Missense mutations of the CD71 gene, *TFRC*, causes immunodeficiency [133] and defective T cell proliferation [134], showing that CD71 expression and the resulting iron flux are important for T cell activation. A recent study showed that in SLE-prone mice and patient T cells, CD71 cell surface expression is increased, resulting from increased endosomal recycling to the plasma membrane. Upon T cell activation, the recycling of CD71 is increased by the fast endocytic sorting pathway controlled by Rab5 and Rab11a [135] (Figure 1). SLE patients with high CD71 expression on their Th17 cells have increased disease severity. In lupus mice, T cells that had increased CD71 expression level also had elevated intracellular iron levels [93]. Blocking CD71 inhibits Th17 differentiation by increasing IL-2 expression and reduces the recruitment of RORγt to the *IL17a* locus [136]. The deletion of *Tfrc* in mouse CD4^+^ T cells showed stronger fitness advantage to Tregs [93]. Interestingly, in Th cells, iron controls pathogenicity by promoting glucose metabolism [137].

### 3.4. NAD^+^ Synthesis and Metabolism

In the last few years, decreased cellular nicotinamide adenine dinucleotide (NAD^+^) levels have been associated with aging [138] and various diseases such as obesity [139], neurodegenerative diseases [140], cancer [141,142], and SLE [143]. NAD^+^ is a metabolite that plays an important role in cellular homeostasis. Cellular NAD^+^ levels are regulated through NAD^+^ metabolism, synthesis, and NAD^+^-dependent non-reduction–oxidation reactions. NAD^+^ is a cofactor in energy metabolism reduction–oxidation (redox) reactions, such as glycolysis, oxidative phosphorylation (OXPHOS), the tricarboxylic acid (TCA) cycle, and fatty acid oxidation [144,145]. NAD^+^ also participates in other non-redox processes, including post-translational modifications [146], mitochondrial metabolism [147], cell signaling [148], inflammatory responses [149], and apoptosis [150]. In non-redox reactions, NAD^+^ is a substrate to enzymes, such as sirtuins [151], poly-ADP-ribose polymerases (PARPs) [152], and the cyclic adenosine diphosphate-ribose (cADPR) family of ectoenzymes [153]. Sirtuin is a family of seven essential histone deacetylases that are involved in several cell signaling regulatory pathways, such as cell survival and longevity [138,151]. PARPs are a family of DNA repair enzymes that mediate a process where NAD^+^ acts as a donor of ADP-ribose moieties [154]. Since NAD^+^ is required for both redox and non-redox processes, cells synthesize NAD^+^ in different pathways.

NAD^+^ synthesis arises from different dietary precursors via (1) the de novo pathway from the amino acid precursor, tryptophan; (2) the Preiss–Handler pathway from nicotinic acid (NA); and (3) the nucleoside pathway from nicotinic acid riboside (NAR) or nicotinamide riboside (NR) [155]. However, the salvage pathway is preferred, where nicotinamide (NAM), the catabolic product of NAD^+^-consuming enzymes (sirtuins, PARPs, and NAD^+^ hydrolases), is recycled for NAD^+^ synthesis [156] (Figure 3). 

#### 3.4.1. Amino Acid Transporter, CD98

In T cells, kynurenine is transported by System L amino acid transporters, which are heterodimers with a heavy chain, SLC3A2 (CD98), and an amino acid transporting light chain, SLC7A5 (LAT1) [157] (Figure 3). In SLE, tryptophan is depleted, while its catabolite, kynurenine, is increased, which may affect the de novo pathway of NAD^+^ synthesis [158,159,160]. CD98 expression was increased in a lupus-prone mouse model [161], so the endosomal recycling pathway that may regulate its expression may play an important role in intracellular NAD^+^ synthesis. After endocytosis, CD98 is recognized and sorted on endosomes by Rab22a and Hook1, a cargo-tethering protein, routing the cargo to recycling endosomes [162] (Figure 1). As mentioned above, in CD4^+^ T cells, the trafficking of LAT1 between the IS and the TGN is controlled by Rab6 and Syntaxin-16 [88]. The accumulation of kynurenine has been attributed to the overgrowth of tryptophan-producing bacteria in the gut microbiota of lupus-prone mice [163,164]. Kynurenine is a most predictive metabolic biomarker in SLE, which triggers mechanistic target of rapamycin (mTOR) activation and is responsive to treatment with NAC [165]. Direct mTOR blockade with sirolimus also reverses inflammatory T cell activation in SLE [10,127,166,167,168].

In glutaminolysis, glutaminase converts glutamine to glutamate. In addition to glycolysis, glutaminolysis also plays an important role in energy production in T cells [111]. Alanine–serine–cysteine transporter 2 (ASCT2, SLC1A5) transports amino acids into the cell, including glutamine (Figure 2). The recycling of glutamine transporters is regulated by retromer, an effector of Rab7 [169], which mediates the transfer of cargo between the endosome and TGN [170]. The deletion of *Slc1a5* impairs Th17 differentiation, confirming the importance of glutaminolysis in Th17 cells [171]. 

#### 3.4.2. NAD^+^ Hydrolases, CD38, CD157, and SARM1

Genes encoding CD38 and its homologue, CD157, are both located on the human chromosome 4 [172,173]. CD38 and CD157 are both receptors and ectoenzymes that have ADP-ribosyl cyclase activity and can convert NAD^+^ into cADPR and hydrolyze cADPR to form ADP-ribose (ADPR), although CD157 is one hundred times less efficient than CD38 (Figure 3) [174,175,176,177]. CD38 was first observed on T cells as an activation marker [178], and now, it is considered ubiquitous in the immune system, with variable levels of expression [176]. CD157 is mainly expressed by myeloid cells, especially neutrophils and monocytes [176,179], and its expression is increased in rheumatoid arthritis patients [180].

CD38 expression levels on T cells have been studied in numerous diseases. CD38^bright^ CD8^+^ T cells can predict acute graft-versus-host disease [181], phenotypic changes in CD38^+^ CD4^+^ T cells can predict the severity of inflammatory bowel disease [182], and CD38 expression in CD4^+^, CD8^+^, or CD25^+^ T cells is significantly higher in SLE patients than in healthy controls [183]. CD38 has a short cytoplasmic tail, and it controls both intra- [177] and extra-cellular [184] NAD^+^ levels. It takes 100 NAD^+^ molecules to generate one cADPR molecule, which indicates that CD38 is a significant NAD^+^ consumer [185,186]. In human CD8^+^ T cells and the Jurkat CD4^+^ T cell line, CD38 decreases NAD^+^ levels [143]. Since the level of CD38 expression on T cells impacts NAD^+^ levels, which is used in a variety of redox and non-redox pathways, determining their functions [187], it is important to determine how its expression is controlled by endosomal recycling. To date, what we know thus far is that in human lymphocytes and T cell leukemia lines, the downregulation of CD38 expression by endocytosis is not a key step in its intracellular signaling; rather, it is a negative feedback mechanism [188]. Upon TCR engagement, CD38 is actively recruited at the immune synapse from the plasma membrane and recycling endosomes [189].

#### 3.4.3. CD73

CD73 is an ecto-5′-nucleotidase that is upregulated in cancer cells [190,191,192], while in SLE patients, it is selectively silenced in B cells and its expression decreased in T cells [193,194,195]. CD73 is suggested to have dual enzymatic functions: (1) cleaving NAD^+^ to NMN and adenosine monophosphate (AMP) and (2) hydrolyzing NMN to NR [196,197,198] (Figure 3). There is an ongoing debate regarding the role of CD73 in contributing to intracellular levels of NAD^+^ in human primary cells [199,200]. Interestingly, CD73^+^ Tregs inhibit CD4^+^ CD25^−^ T cell proliferation via the Treg-derived exosomes from late endosomes [201,202] (Figure 1). The expression of CD73 in these CD73^+^ Treg-derived exosomes [202] suggests the need for further study to determine its endosomal pathway.

In the small intestine, SLC12A8, an NMN transporter, is highly expressed, which suggests that NMN may be an entry point into the nucleoside NAD^+^ synthesis pathway [203]. It is also overexpressed on bladder cancer cells and associated with tumor immune cell infiltration [204]. However, its expression and mechanisms of NMN uptake in immune cells remain unclear.

### 3.5. Role of Endosome Traffic in Toll-like Receptor-Mediated Signaling

Toll-like receptors (TLRs) are classified by the location of their expression: intracellular or extracellular. TLR2 and TLR4 are expressed on the surface of CD4^+^ T cells [205], and when they bind to their ligand, i.e., bacterial membrane components [206,207], they are endocytosed to the endosomes, a step required for the activation of nuclear factor-κB (NF-κB) and AP-1 [208,209]. In endosomes, Rab11a promotes TLR4 signaling [210], and Rab10 traffics TLR4 back to the cell surface [211]. Recently, it has been shown that Rab8 and Rab11 recruits adaptor protein-3 (AP-3) to TLR2, promoting the secretion of IL-6 by phagocytic cells [212]. TLR2 expression level is increased in SLE patients’ CD4^+^ T cells, CD8^+^ T cells, and B cells [213]. TLR2 activation in human Tregs reduces their suppressive functions, and it promotes Th17 differentiation in naïve CD4^+^ T cells [214]. On the other hand, intracellular TLR7 and TLR8 are confined to endosomes, where they sense viral single-stranded RNAs [215]. Genetic variants of TLR7 and TLR8 are risk factors for SLE [216,217], and their expression is upregulated in SLE patients [218]. TLR7 activation in CD4^+^ T cells induces calcium flux [219] and TLR8 signaling in human Tregs inhibits glucose uptake and glycolysis [220].

## 4. Proinflammatory Signaling Pathways Impacted by Trafficked Receptors in CD4^+^ T Cells

SLE results from multiple predisposing genetic traits and environmental stimuli [21]. Environment stimuli can induce alterations to membrane trafficking to increase or decrease surface protein expression [26]. The changes in surface receptor expression of a cell affects how the cell detects stimuli, such as cytokines from other immune cells, and the downstream effects of how it responds by producing other cytokines. A defect in the cell’s response, such as an uncontrolled cytokine production, can lead to immune activation and tissue damage in SLE, and aberrant signaling pathways contribute to this phenomenon [221]. Many studies have revealed that metabolism is important in T cell differentiation and function [111]. We will discuss how the above discussed changes in metabolites resulting from surface receptor recycling may lead to proinflammatory and anti-inflammatory imbalance in SLE (Table 1).

### 4.1. IL-2 and Tregs Are Decreased

Cytokine abnormalities are found in the pathogenesis of SLE [21]. IL-2 is produced by activated Th cells and plays an important role in expansion and homeostasis [21]. IL-2 production is decreased in the T cells of SLE patients and contributes to the Th17/Treg imbalance in SLE [222]. Naïve CD4^+^ T cells can be skewed to become Tregs when their TCR is stimulated in the presence of IL-2 and TGFβ [223]. The expansion of Tregs is greatly dependent on IL-2, which suggests that IL-2 depletion contributes to the reduced number and function of Tregs in SLE patients [224]. Tregs control the expansion of autoreactive T cells, and thus are important in inhibiting autoimmunity [13]. Even though it is well established that SLE patients have defective IL-2 production, the underlying mechanism is poorly understood.

#### 4.1.1. CREM and CREB Control IL-2 Production

Several transcription factors control *IL2* transcription, but the most well described in SLE patients is the imbalance between cyclic AMP responsive element-binding protein (CREB) and cyclic AMP element modulator (CREM). CREB is a positive regulator and CREM is a negative regulator, and they compete to bind to the *IL2* promoter. In resting T cells, CREB is bound, and when the T cell is activated, CREB is phosphorylated, leading to *IL2* transcription (Figure 2). However, when phosphorylated, CREB is replaced with phosphorylated CREM, and *IL2* transcription is repressed [225]. 

In SLE patients’ T cells, CREB is decreased, while CREM is abnormally increased [225]. Furthermore, there is evidence that calcium/calmodulin-dependent protein kinase IV (CaMKIV), an enzyme that phosphorylates CREM, increases the binding of the repressor CREM to the *IL2* promoter, suppressing *IL2* transcription [13]. Circulating autoantibodies and autoantigens common in SLE patients, such as anti-TCR antibody, can also activate CaMKIV [226]. Moreover, SLE T cells demonstrate hypomethylation of protein phosphatase 2 (*PP2A*) promoter, increasing the levels of serine/threonine phosphatase, PP2A [227], which binds to CaMKIV and keeps it catalytically inactive in the cytoplasm [228]. Increased calcium flux in SLE T cells promotes the accumulation of Ca^2+^/calmodulin (CaM), which replaces PP2A from CaMKIV, activating it. This inhibition of CaMKIV/PP2A results in increased CaMKIV-mediated gene transcription [228]. PP2A is responsible for the dephosphorylation of CREB [229], and it also dephosphorylates and activates SP-1, which then promotes *CREM* transcription [230], ultimately decreasing IL-2 production (Figure 2). *CREM* transcription is increased by the abnormally increased amounts of activated transcription factor SP-1 shown in SLE [230].

#### 4.1.2. NFAT and AP-1 Control IL-2 Production

In SLE T cells, there is an increase in calcium influx [231]. The increase in cytosolic calcium concentration leads to increased calcineurin activation. Activated calcineurin dephosphorylates the nuclear factor of activated T cells (NFAT) in the cytoplasm, which is abnormally high in SLE [232]. The dephosphorylated NFAT then translocates to the nucleus, where it would normally bind to the promoters of *IL2* genes [233]. Calcium signaling and NFAT activity are reinforced by aerobic glycolysis-derived PEP [119] (Figure 2). However, in SLE, NFAT does not promote IL-2 production because the *IL2* promoter requires the binding of a transcription factor AP-1 to adjacent sites [232].

The AP-1 transcription factor family is formed by heterodimers and homodimers of Fos and Jun proteins [234]. When TCR binds to an antigen, the Fos and Jun proteins are expressed, and AP-1 binds to the *IL2* promoter (Figure 2). However, c-Fos expression is decreased in SLE T cells, which reduces AP-1 binding to the *IL2* promoter [235]. c-Fos contains cyclic AMP responsive element (CRE) sites in its promoter and CREM downregulates c-Fos activity [236]. Since CREM is increased in SLE T cells, CREM binds to the c-Fos promoter and decreases c-Fos production [237]. Therefore, even though NFAT expression is increased in SLE, due to the reduced AP-1 activity, IL-2 production is still reduced [232]. 

The sirtuins remove acetyl groups from transcription factors and histones, inhibiting gene transcription [238]. In CD4^+^ T cells, sirtuin-2 deacetylates c-Jun and histones at the *IL2* gene, decreasing IL-2 production [239] (Figure 3). In experimental autoimmune encephalomyelitis (EAE) mice and lupus-prone mice, a sirtuin-2 inhibitor, AK-7, ameliorated disease severity [239]. Since sirtuins are NAD^+^-dependent histone deacetylases, this suggests that NAD^+^-modulating cell surface molecules may play an important role. 

In SLE T cells, blocking CD71 with an antibody normalizes T cell activation and IL-2 production [93,240]. However, in a Jurkat CD4^+^ T cell line, blocking CD71 with an antibody did not change the activation profile of NFAT, AP-1, and nuclear factor kappa-light-enhancer of activated B cells (NF-κB) [240]. Further study is needed to investigate the pathways through which CD71 affects IL-2 production. 

When TCR is stimulated, CRACR2A transmits the signal to activate the Ca^2+^-NFAT and JNK-AP1 pathways [108]. As mentioned above, biallelic CRACR2A mutations cause autoimmune inflammatory complications, and STIM1, which CRACR2A traffics, is significantly increased in lupus mice kidneys and contributes to renal damage [108,241]. In a Jurkat CD4^+^ T cell line, Loureirin B, a constituent from a traditional Chinese medicine, modulates IL-2 secretion by inhibiting STIM1/ORAI1 channels and decreasing calcium influx [242]. In SLE patients’ T cells, forcing CD3ζ expression returns calcium fluxing and IL-2 production to normal, which indicates that these phenomena are downstream effects of altered calcium signaling [243]. This decrease in IL-2 production has effects on other cytokine and lineage development, such as hindering *IL17a* expression and Th17 differentiation. This may explain the increased IL-17 production in SLE [244,245]. 

### 4.2. IL-17 and Th17 Are Increased

SLE patients have increased serum levels of IL-17 [246] and an increased frequency of IL-17-producing cells in the peripheral blood [247,248,249]. IL-17 is pro-inflammatory because it induces IL-6, granulocyte monocyte-colony stimulating factor (GM-CSF), and granulocyte colony stimulating factor (G-CSF) production [250,251,252,253,254]. This recruits monocytes and neutrophils, leading to inflammation and tissue damage. Furthermore, in the presence of the B cell activating factor, IL-17 increases B cell activation, proliferation, and differentiation into immunoglobulin-secreting cells [255]. Th17 cells play an important role in the pathogenesis of SLE by amplifying inflammation [256].

IL-17 is produced rapidly and in large amounts by γδT cells, DN TCRαβ T cells, and Th17 cells [247,257,258,259]. DN T cells are increased and represent a major source of IL-17 in SLE patients [247]. TCRαβ DN T cells derive from CD8 T cells [260], and their effector functions play a key role in tissue damage seen in SLE. They also produce other pro-inflammatory mediators, namely interferon (IFN)-γ, IL-1β, CXCL2, and CXCL3, and infiltrate the kidneys, the site of lupus nephritis, one of the leading causes of death in SLE [247,260,261]. On the other hand, Th17 cells derive from naïve CD4^+^ T cells primed in the presence of TGFβ, IL-6, IL-21, and IL-1β [20,262,263]. IL-23, which is produced by antigen-presenting cells, can also induce the expansion of Th17 cells [19].

#### 4.2.1. JAK/STAT3 Pathway Regulates Th17 Differentiation

Calcium influx is increased in SLE T cells due to the loss of TCRζ [100]. The increase in cytosolic calcium concentration leads to increased calcineurin activation. Activated calcineurin dephosphorylates inactive NFAT, which is expressed at abnormally high levels in SLE T cells [232]. The dephosphorylated NFAT then translocates to the nucleus, where it binds and activates the promoters of CD40 ligand (CD40L) [233]. CD40-CD40L signaling plays an important role in the differentiation of Th17 cells [264].

CD40 is expressed by APCs and CD40L is on T cells. Various methylation-sensitive genes that functionally contribute to SLE pathogenesis are overexpressed in SLE patients’ CD4^+^ T cells, including CD11A, CD70, and CD40L [265,266]. Upon exposure to high antigen doses, T cells upregulate CD40L, providing efficient co-stimulation to APCs to produce IL-6 [264], which is elevated in SLE [18]. IL-6 is produced in many cell types, like monocytes and B cells [267]. IL-6 signaling induces Th17-related gene expression via the signal transducer and activator of transcription (STAT)3 [268].

Most cells respond to IL-6 trans-signaling, which is achieved through IL-6 binding to soluble IL-6 receptor (sIL-6R), rather than the classic signaling via IL-6R on the cell’s surface [269]. In human peripheral blood mononuclear cells, the cleavage of IL-6R to generate sIL-6R is induced by the activation of TLR2 [270]. When IL-6 binds to IL-6R, the IL-6R subunit-β (gp130; IL-6Rβ) initiates intracellular signaling and activates gp130-associated Janus kinases (JAKs) [271]. The activated JAKs phosphorylate gp130, initiating the JAK/STAT3 pathway [271]. The SRC homology domain 2 (SH2) domain of STAT3 binds to the gp130 phosphotyrosine docking sites. When STAT3 is within proximity of active JAKs, STAT3 is phosphorylated at Tyr705, resulting in the dimerization of the STAT3 protein and nuclear translocation [272]. STAT3 binds to the promoters of target Th17-related genes, such as *IL17a*, *IL17f*, and *IL23r* [273] (Figure 2). IL-23 is critical for Th17 maintenance [274]. Therefore, increased calcium influx due to Rab4A-mediated lysosomal degradation of TCRζ in SLE T cells leads to the activation of NFAT, which upregulates CD40L expression. CD40L stimulation leads APCs to produce IL-6 [264], which leads to the activation of JAK/STAT3 pathway in the T cell [271], causing an increase in IL-17 production and Th17 development [264].

#### 4.2.2. mTOR Regulates IL-17 Production and Th17 Differentiation

CaMKIV also plays an important role in IL-17 production and Th17 development using two pathways: (1) increasing the binding of CREMα to *IL17* genes and (2) via the AKT/mTOR pathway. CaMKIV activity is the highest under Th17 polarizing conditions. In mice, when CaMKIV is inhibited, Th17 and Treg differentiation is affected but not Th1 or Th2 [275]. 

CREM plays an important role in IL-2 production, but it also plays a role in the Th17 differentiation and IL-17 production through epigenetic remodeling [276]. The suppressor isoform, CREMα, is increased in SLE T cells and controls IL-17A expression by reducing DNA methylation of the *IL17A* locus [277]. DNA methylation is when methyl groups are added to DNA molecules, and in eukaryotes, cytosine is methylated at the 5-carbon [278]. DNA methylation occurs at CpG sites, the regions of DNA where cytosine is adjacent to a guanine nucleotide reading in the 5′ to 3′ direction. The addition of methyl groups represses gene expression [279]. Therefore, the reduction in CpG-DNA methylation by CREMα lifts the repression of gene expression, leading to increased *IL17* transcription [277].

Furthermore, CaMKIV also promotes Th17 differentiation and IL-17 production via the PI3K/AKT/mTOR pathway [275] (Figure 2). The PI3K/AKT/mTOR pathway plays an important role in proliferation, growth, and survival [280]. mTOR is a component of two complexes, mTOR complex 1 (mTORC1) and mTOR complex 2 (mTORC2). A key function of mTORC1 is its ability to phosphorylate key regulators of mRNA translation [281], while a key function of mTORC2 is the phosphorylation and activation of serine/threonine protein kinase B (PKB; AKT) through phosphorylation at Ser473 [282]. A key effector downstream of AKT is mTORC1 [281]. The activation of mTORC1 enhances Th17 differentiation [283] and the disruption of mTORC1 impairs Th17 differentiation [284]. The TCR signal triggers mTORC1 activation by inducing System L amino acid transporters to uptake leucine [285]. Rapamycin inhibits mTORC1 but not mTORC2. Rapamycin has shown to normalize TCR-induced calcium influx [286] and Th17/Treg balance [287] in SLE patients.

In SLE, FcεRIγ replaces CD3ζ [96], resulting in the recruitment of Syk instead of ZAP-70 [59]. Syk has been shown to activate the PI3K/AKT/mTOR pathway. Active Syk increases the catalytic activity of PI3K, resulting in the conversion of PIP2 to phosphatidylinositol-3,4,5-triphosphate (PIP3) [288]. This then recruits phosphatidylinositol-dependent kinase 1 (PDK1) and AKT to the cell membrane [289]. Then, PDK1 is phosphorylated at Thr308 in the AKT kinase catalytic region [290], and mTORC2 phosphorylates AKT [276]. Activated AKT then phosphorylates tumor suppressor TSC2, leading to mTORC1 activation [291] (Figure 2). In SLE T cells, mTORC1 activity is increased, while mTORC2 is reduced [292]. 

mTORC1 phosphorylates the ribosomal S6 kinases (S6K) [293]. There are two homologs of S6K: S6K1 and S6K2. More research has been conducted on S6K1 than S6K2 due to the belief that their high degree of homology leads them to behave similarly [294] (Figure 2). However, S6K1 and S6K2 lead to different pathways that both positively regulate Th17 differentiation [283].

The PI3K/AKT/mTORC1/S6K1 axis promotes Th17 differentiation via growth factor-independent 1 transcriptional repressor (Gfi1) [283]. Gfi1 is a transcriptional repressor that recruits histone-modifying enzymes to the target gene promoters [295]. The downregulation of Gfi1 is a critical event for Th17 differentiation [296]. The PI3K/AKT/mTORC1/S6K1 axis induces early growth response 2 (EGR2), a transcription regulatory factor that contains zinc finger DNA-binding sites, and its increased gene expression is a risk factor for SLE [283,297]. EGR2 downregulates the expression of Gfi1 and increases Th17 differentiation [283] (Figure 2).

On the other hand, the PI3K/AKT/mTORC1/S6K2 axis promotes the nuclear translocation of RORγt [283], a critical transcription factor for the initiation of Th17 differentiation [298]. S6K2 interacts directly with RORγt and enhances its nuclear translocation [283]. The PI3K/AKT/mTORC1 pathway enhances the expression of S6K2, accelerating RORγt nuclear translocation during Th17 differentiation [283] (Figure 2). 

Pyruvate kinase muscle isozyme 2 (PKM2) acts at the last step of glycolysis, and it is required for Th17 differentiation [299]. It translocates to the nucleus and activates with STAT3, increasing Th17 differentiation [300]. PKM2 also binds to CaMKIV, and CaMKIV enhances pyruvate kinase activity and glycolysis through the glucose transporter [301] (Figure 2).

CREM also enhances RORγt. Inducible cyclic AMP early repressor (ICER) is a transcriptional repressor isoform of CREM, and it enhances the accumulation of RORγt and binds to the *IL17a* promoter, leading to IL-17 production [302]. ICER also induces Th17 differentiation via the sirtuin-2/mTORC1/HIF-1α pathway [239]. Following TCR signaling and mTORC1, hypoxia-inducible factor 1-alpha (HIF-1α) is induced, which is normally overexpressed in Th17 cells [303]. ICER binds to the *Sirt2* promoter, sirtuin-2 deacetylates p70S6K, activating the mTORC1/HIF-1α/RORγt pathway and inducing Th17 differentiation [239] (Figure 3). HIF-1α also upregulates GLUT1, promoting increased glucose uptake, and upregulates PDK1, reinforcing glycolysis [304] (Figure 2).

The STAT3 phosphorylation and activation is not limited to JAKs [271]. It was generally believed that JAK/STAT3 signaling pathway contributed to STAT3 activation through tyrosine and serine phosphorylation. However, studies have suggested that STAT3 can be activated through Ser727 phosphorylation in the absence of Tyr705 phosphorylation, the site of phosphorylation by JAKs [305]. Tyr705 phosphorylation by JAKs is involved in STAT3 dimerization and activation [306], while Ser727 phosphorylation modulates STAT3 activity [307,308]. mTOR phosphorylates STAT3 on Ser727 [309].

As discussed above, in SLE T cells, mTORC1 activity is increased while mTORC2 is reduced [292]. Rapamycin has shown to reduce STAT3 activation and the number of IL-17 producing cells in SLE patients [310]. Therefore, the increase in calcium flux in SLE T cells due to the lysosomal degradation of CD3ζ can lead to an increase in Th17 development through two pathways: S6K2/RORγt pathway and STAT3 pathway (Figure 2).

#### 4.2.3. Histone Modification Regulates Th17 Differentiation and IL-17 Production

In mice, the loss of sirtuin-1 functions results in increased T cell activation and a lupus-like phenotype [311]. One of the targets for sirtuin-1 is STAT3. In addition to phosphorylation, acetylation on Lys685 regulates the transcriptional activity of STAT3, and this can be inhibited by sirtuin-1 [312,313]. Sirtuin-1 deacetylates STAT3, reducing its ability to translocate to the nucleus [314]. In cancer cell lines, decreased NAD^+^ activates STAT3 [315]. As mentioned above, sirtuin-2 deacetylates p70S6K, inducing Th17 differentiation via the mTORC1/HIF-1α/RORγt pathway [239] (Figure 3). Therefore, NAD^+^ levels may have an important role in STAT3 activity by targeting Th17-related genes, suggesting the importance of surface molecules that modulate intracellular NAD^+^ levels. 

TLR2 activation in CD4^+^ T cells leads to pro-inflammatory skewing [214]. In vitro stimulation of TLR2 on SLE patient CD4^+^ T cells with a synthetic bacterial lipopeptide, Pam3-Cys-Ser-Lys4 (Pam_3_CSK_4_), significantly induces IL-17A and IL-17F production by upregulating H3K4 tri-methylation levels in the IL-17A promoter region and H4 acetylation levels in both IL-17A and IL-17F promoter regions [213]. 

### 4.3. Regulation of Tfh Development

DNA demethylation plays an important role in CD4^+^ T cell differentiation [316,317]. The methylome of SLE patients’ CD4^+^ T cells differ from those of healthy controls [318]. PP2A inhibits DNA methyltransferase 1 (DNMT1) expression [319]. Under the presence of Fe^2+^, the ten-eleven translocation (TET) enzymes oxidize 5-methylcytosine to 5-hydroxymethylcytosine in DNA, hypomethylating DNA and controlling gene transcription [320] (Figure 2). Therefore, genes involved in SLE pathogenesis may undergo hypomethylation, becoming upregulated [111]. 

SLE is about nine times more common in women than men in the United States [321]. However, SLE tends to be more severe in men with a higher prevalence of renal disease [322]. To achieve an SLE disease flare with a severity equal to women, men require a higher genetic risk and a greater degree of CD4^+^ T cell demethylation [323]. Compared to female SLE patients, male patients’ CD4^+^ T cells show significant hypomethylation of *ELAVL1*, *UHRF1*, and *SMAD2*, increasing their gene expressions [324]. Embryonic lethal vision-like protein 1 (ELAVL1) is an RNA-binding protein (RBP) that interacts with immune-specific transcripts, influencing the intensity of the immune response [325,326,327]. ELAVL1 stabilizes the mRNA of Ubiquitin-like with PHD and RING finger domains 1 (UHRF1), augmenting its effects [324]. UHRF1 regulates DNA methylation during the S phase of the cell cycle [328]; therefore, increased UHRF1 expression could lead to increased CD4^+^ T cell proliferation in male SLE patients [324]. UHRF1 also promotes Treg proliferation by hypermethylating p21 [329] and reduces follicular helper T cell (Tfh) differentiation [330]. Thus, UHRF1 may play a mixed role in SLE pathogenesis in male SLE patients [324].

Pyruvate, the product of glycolysis, can be broken down into acetyl-CoA by pyruvate dehydrogenase (PDH) (Figure 2). Histone acetylases transfer the acetyl group from acetyl-CoA to histones [331]. SLE patients’ CD4^+^ T cells have hypoacetylated histones [265,332], and treating lupus-prone mice with histone deacetylase inhibitors alleviated disease severity by downregulating proinflammatory cytokines [333].

In vitro, IL-6 and IL-21, which act through STAT3, drive Tfh differentiation [334]; however, in vivo, IL-6 and IL-21 are not absolutely required for Tfh development [335], suggesting the role of other cytokines in Tfh development. Type I IFN activates STAT1 to bind to B cell lymphoma 6 (Bcl6), a transcription factor that is required for Tfh development [336], and in human and SLE mouse model, type I IFN signaling activates STAT4 to produce IL-21 and IFNγ [337]. In HeLa cells, Rab7 is required for early endosomal sorting of the subunit 1 of the type I IFN receptor (IFNAR1) [338], and recently, anifrolumab, a monoclonal antibody targeting IFNAR1, has been FDA-approved for SLE [339]. Therefore, endosome traffic of IFNAR1 in CD4^+^ T cells may modulate the clinical efficacy of this new therapeutic intervention in SLE.

**Table 1 ijms-24-10749-t001:** Receptors that are recycled by Rab GTPase, downstream metabolites, and subsequent signaling/epigenetic regulation on Treg/Th17 differentiation. Expression level in SLE T cells is indicated in parentheses, if known. ↑ = increased in SLE; ↓ = decreased in SLE; ? = Unknown.

Recycled Receptors	Responsible Rab GTPase	Downstream Metabolites	Subsequent Signaling/Epigenetic Pathways	Effects on T Cell Subset/Cytokine	References
CD3ζ (↓)	Rab4A (↑)	Ca^2+^ (↑)	Ca^2+^/CaM/PP2A/de-pCREB	Decrease IL-2 (↓)	[10,229]
			Ca^2+^/CaM/PP2A/SP-1/CREM (↑)		[10,230,231]
			Calcineurin/NFAT (↑) (no AP-1)		[174,175,176,177,232]
			CaMKIV/CREMα	Increase IL-17 (↑) and Th17 (↑)	[10,277]
			CaMKIV/PI3K/AKT/mTORC1 (S6K)		[10,275]
			Syk/PI3K/PIP3/PDK1/AKT/pTSC2/mTORC1 (S6K)		[10,59,283,288,289,291]
CD38(↑)/ CD157	?		CREM/ICER/RORγt		[174,175,176,177,302]
			ICER/Sirtuin-2/mTORC1/HIF-1α/RORγt		[174,175,176,177,303]
			CREM/ICER/Sirtuin-2/p70S6K/HIF-1α/RORγt		[174,175,176,177,239,255]
		NAD^+^ (↓)	Sirtuin-1/STAT3	Increase Th17	[174,175,176,177,312,313]
			Sirtuin-2/de-Ac c-Jun	Decrease IL-2	[157,162,239]
CD98 (↑)	Rab22a	Kynurenine	NAD^+^ synthesis	-	[158,159,160]
CD4 (↓)	Rab4A (↑)	?	?	?	[61]
CD71 (TfR) (↑)	Rab4A, Rab5, Rab11a	Fe^2+^	TET/DNA hypomethylation	Tfh	[61,93,135,320]
GLUT1 (↑)	Rab25	Glucose	PEP/Ca^2+^/NFAT (no AP-1)	Decrease IL-2 (↓)	[117,119,232]
			PMK2/STAT3	Increase Th17 (↑)	[299,300]

## 5. Conclusions

The factors that mediate SLE pathogenesis and organ damage span the whole immune system. Therefore, it is not surprising that the cytokine networks that orchestrate immune cell functions are dysregulated. The dysregulation of IL-2 is fundamental for SLE pathogenesis, as it reduces Treg differentiation, disturbing immunologic homeostasis [13]. The IL-17 signature plays an important role in tissue damage mediated by effector cells, as it recruits other pro-inflammatory cells and increases B cell activation and antibody production [255].

mTOR is a sensor of the mitochondrial transmembrane potential, which is increased in SLE T cells [10]. The activation of mTOR leads to the overexpression of the Rab5A and Rab4A small GTPases, which regulate the endocytic recycling of surface receptors, as described above [10]. Rab4A-dependent lysosomal degradation contributes to the loss of TCRζ and CD4 in SLE T cells [10], causing changes in the calcium signaling pathway compared to normal T cells. The dysregulated TCR signaling in SLE leads to a decrease in IL-2 production and increases in IL-6 and IL-17 production, causing a decrease in Treg differentiation and an increase in Th17 differentiation. GLUT1 also plays a role in activating the calcium signaling pathway through generating PEP during glycolysis [119].

Upon TCR engagement, CD38 is recycled to the immune synapse from the recycling endosomes [189]. CD38 and its homologue, CD157, convert NAD^+^ into cADPR, controlling both intra- [177] and extra-cellular [184] NAD^+^ levels. The intracellular NAD^+^ levels control the activity of sirtuins, histone deacetylases that inhibit gene transcription by removing acetyl groups from histones and transcription factors [238]. Sirtuin-1 deacetylates STAT3, reducing its nuclear translocation [314]. Sirtuin-2 deacetylates c-Jun and decreases IL-2 production while activating the mTORC1/HIF-1α/RORγt pathway and inducing Th17 differentiation [239]. mTOR regulates iron homeostasis by modulating CD71 stability [340]. The recycling of CD71 is increased upon T cell activation [135], and on SLE patients’ Th17 cells, elevated CD71 expression has been correlated with disease severity [93]. Furthermore, TET requires Fe^2+^ to hypomethylate DNA and control gene transcription, affecting Tfh cell differentiation [320].

This correlation between receptor recycling, mTOR activation, metabolic pathways, and T cell reactivity warrants further study on how Rab GTPases ultimately contribute to the pathogenesis of SLE.

## Figures and Tables

**Figure 1 ijms-24-10749-f001:**
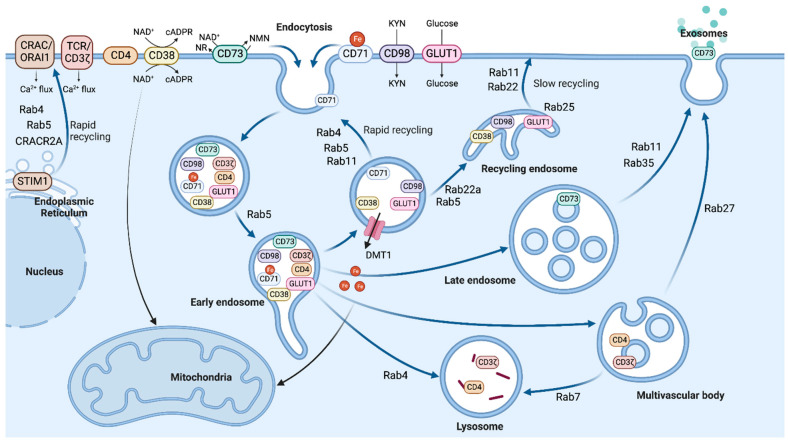
Endosomal trafficking and recycling pathways are illustrated. Cell membrane proteins are endocytosed, then the internalized cargoes converge into a common early endosome. There are two pathways of recycling back to the plasma membrane: (1) the rapid-recycling pathway, directly from the early endosome, and (2) the slow-recycling pathway, via the recycling endosome. The cargoes can be released out of the cell as exosomes through the late endosome or the multi-vesicular body. Some cargoes from endosomes are routed to the lysosome for degradation. Rab GTPases play important roles in receptor recycling, including endocytosis, intracellular vesicle trafficking, and exosome secretion. Key Rab family proteins are highlighted.

**Figure 2 ijms-24-10749-f002:**
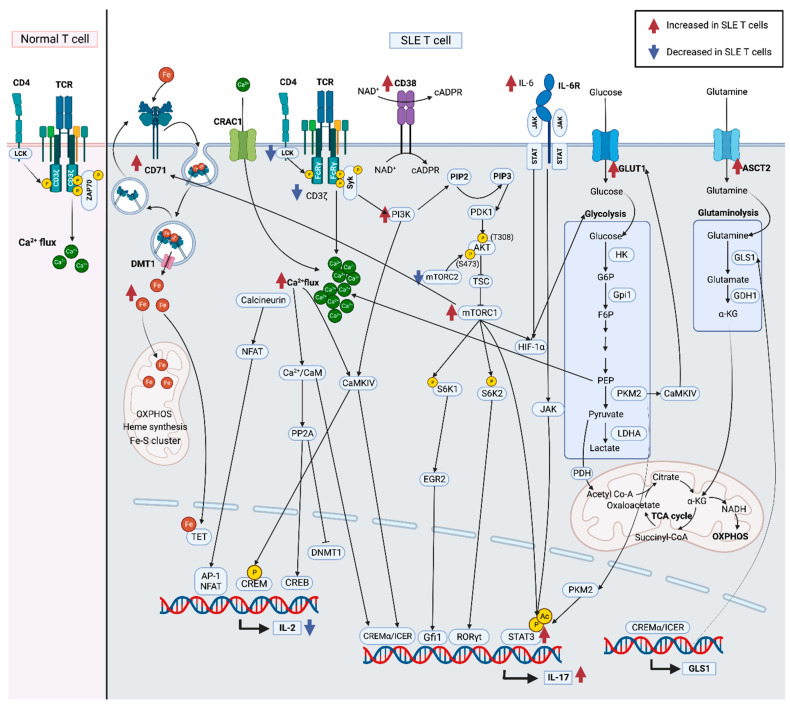
Metabolic abnormalities in SLE T cells and how they contribute to the signaling pathways that control IL-2 and IL-17 production. CD71, which controls the iron flux, is increased in SLE T cells. In place of the CD3ζ protein, FcεRIγ is substituted. In SLE T cells, FcεRIγ recruits Syk instead of ZAP-70. FcεRIγ–Syk interaction is significantly stronger than CD3ζ–ZAP-70 interaction, resulting in higher calcium influx. CD38, which is increased in SLE T cells, converts NAD^+^ into cADPR, which mediates calcium-mobilizing activity. GLUT1, a glucose transporter, and ASCT2, a glutamine transporter, are increased in SLE T cells, and the resulting glycolysis, glutaminolysis, and the TCA cycle are shown. The effects of these metabolites on pathways that affect IL-2 and IL-17 production are shown. Red arrows, increased in SLE T cells. Blue arrows, decreased in SLE T cells.

**Figure 3 ijms-24-10749-f003:**
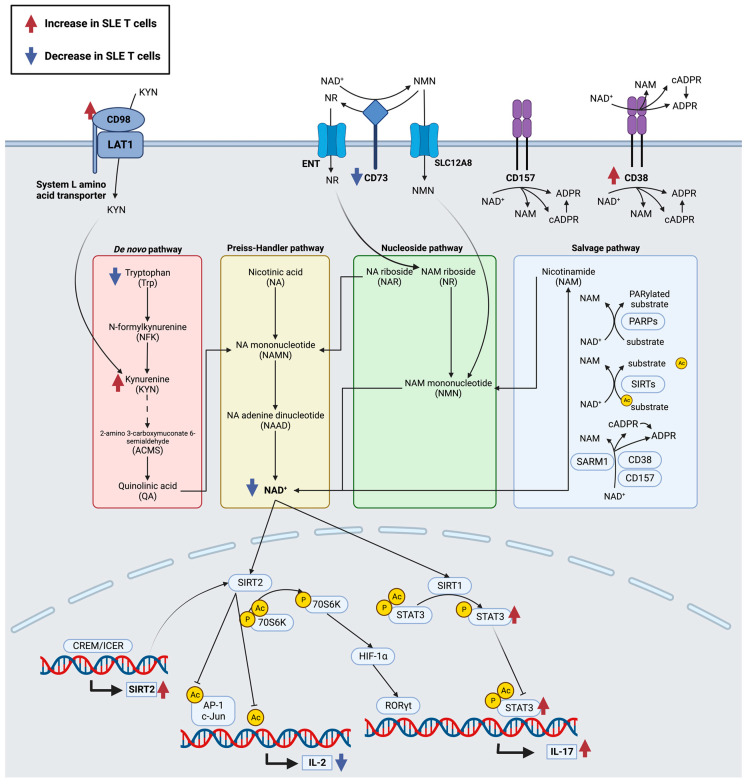
Receptors that contribute to NAD^+^ synthesis pathways in T cells and how NAD^+^ affects IL-2 and IL-17 production. NAD^+^ synthesis pathways shown are: (1) de novo pathway, (2) Preiss–Handler pathway, (3) nucleoside pathway, and (4) salvage pathway. The kynurenine transported by System L amino acid transporters, which are heterodimers of CD98 and LAT1 contribute to the de novo pathway. CD73 has dual enzymatic functions, (1) cleaving NAD^+^ to NMN and (2) hydrolyzing NMN to NR, which both contribute to the nucleoside pathway after being imported into the cell by the nicotinamide mononucleotide transporter, Slc12a8 and equilibrative nucleoside transporter (ENT), respectively. CD38 and CD157 are ADP-ribosyl cyclases, converting NAD^+^ into cADPR and hydrolyzing cADPR to form ADPR. The NAD-dependent histone deacetylases, sirtuins, and their effects on IL-2 and IL-17 production are shown. Red arrows, increased in SLE T cells. Blue arrows, decreased in SLE T cells.

## Data Availability

Not applicable.

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
