# Peer review of "Endosome Traffic Modulates Pro-Inflammatory Signal Transduction in CD4+ T Cells—Implications for the Pathogenesis of Systemic Lupus Erythematosus"

_ijms, 2023, doi:10.3390/ijms241310749_

Round 1
Reviewer 1 Report (New Reviewer)
Park and Perl have reviewed endocytic cycling and related defects in patients with SLE. In my opinion, the review is unsuitable for publication in its current form. Problems include:
1. Inadequate focus on the question of endocytic recycling. If the authors were to focus on endocytic recycling, and any evidence that defects might contribute to the pathogenesis of SLE, this would be a substantial task. They have not confined themselves to these questions For example, section 4.1.1. is a discussion of IL-2 transcriptional regulation. Section 4.2 concerns IL-17 in SLE. As a result of this lack of focus, it is difficult to identify any lessons regarding endocytosis.
2. Uneven level of detail. For example, the very first paragraph describes Rab GTPases, while the next paragraph summarises cytokines. I think if the aims of this review are to be achieved, the authors should assume that their readers possess a certain level of immunological knowledge (e.g. what a cytokine is, how T cells are activated etc) so they can focus their attention on aspects of endocytic cycling function.
3. For a review, the article relies far too heavily on other reviews rather than primary literature Assessment of the relevance of defects recorded in patients with SLE is particularly uncritical. SLE is heterogenous and many patients are receiving immunosuppressive medications. Many mouse models exist, but these vary in the fidelity with which they model human disease. If the review is to advance our understanding, it should identify those findings that offer true mechanistic insights. In other words, where defects in the endocytic pathway are responsible for driving pathology rather than simple associations and correlations.
See for example, line 532:
Naïve CD4+ T cells can be skewed to become Tregs when their TCR is stimulated in the presence 532 of IL-2 and TGFβ219. The expansion of Tregs is greatly dependent on IL-2, which explains the 533 reduced number of Tregs in SLE patients220.
Treg abundance might be dependent on IL-2, and Tregs could be reduced in some patients with SLE, but this does not necessarily mean that IL-2 deficiency is the explanation.
4. Sections ostensibly concerned with outlining the basic mechanisms and function of endosomal recycling are conflated with aspects of disease pathogenesis. This might be unavoidable, but it means that the discussion of disease mechanisms is really far too brief for the reader to understand the mechanisms being discussed. See, for example, the telegraphic description of Sjogren’s syndrome, heart disease (lines 270-274).
5. The structure of the review is disorganised, which makes it hard to read or to understand. For example, Section 2.3 is ostensibly concerned with immune synapse formation. Indeed this section provides a brief overview of this complex topic. Section 3.1, which is under the topic of metabolic programming of T cells, discusses the immune synapse again, as well as the structure of the CD3 complex, and calcium flux. The possible relevance of this discussion is finally revealed at the end of section 3.1.1, where the authors discuss endocytic recycling of STIM1. This point should have been introduced much earlier to provide the reader with the context for the relevance of calcium flux.
Similarly, we have a long discussion about NAD hydrolases (Line 494). It is unclear why this discussion is included because at the end of the section, the authors report:
Since CD38 expression level on T cells impacts NAD+ levels, which is 494 used in a variety of redox and non-redox pathways, determining their functions199, determining 495 how its expression is controlled by endosomal recycling would be important. To date, what we 496 know thus far is that in human lymphocytes and T cell leukemia lines, the downregulation of CD38 497 expression by endocytosis is not a key step in its intracellular signaling, rather it is a negative 498 feedback mechanism200.
6. It is surprising that so little attention is given to endosomal TLR receptors, ligations of which is known to be important in driving lupus pathogenesis.
7. There are countless errors of syntax. For example,
Line 248
“Rab35 is an regulates 248 rapid recycling of CD71 by localizing to the plasma membrane and early endosome58”.
8. Several sections remain inexplicably highlighted.
See above. The problems are not so much with the use of English, but failure carefully proofread the manuscript before submission
Author Response
Reviewer 1
Comments and Suggestions for Authors
Park and Perl have reviewed endocytic cycling and related defects in patients with SLE. In my opinion, the review is unsuitable for publication in its current form. Problems include:
- Inadequate focus on the question of endocytic recycling. If the authors were to focus on endocytic recycling, and any evidence that defects might contribute to the pathogenesis of SLE, this would be a substantial task. They have not confined themselves to these questions For example, section 4.1.1. is a discussion of IL-2 transcriptional regulation. Section 4.2 concerns IL-17 in SLE. As a result of this lack of focus, it is difficult to identify any lessons regarding endocytosis.
Response: In accordance with the reviewer’s comments, we have extended the discussion of endosomal trafficking in Section 1. We provided detailed description of the role specific trafficked receptors in Section 3 and how they contribute to the pro-inflammatory/anti-inflammatory imbalance in SLE in Section 4. We highlighted IL-2 and IL-17 as key cytokines in the pro-inflammatory/anti-inflammatory imbalance in autoimmunity in general and in SLE, in particular.
- Uneven level of detail. For example, the very first paragraph describes Rab GTPases, while the next paragraph summarises cytokines. I think if the aims of this review are to be achieved, the authors should assume that their readers possess a certain level of immunological knowledge (e.g. what a cytokine is, how T cells are activated etc) so they can focus their attention on aspects of endocytic cycling function.
Response: In agreement with the reviewer’s suggestion, we removed the general introduction about cytokines in paragraph 2.
- For a review, the article relies far too heavily on other reviews rather than primary literature Assessment of the relevance of defects recorded in patients with SLE is particularly uncritical. SLE is heterogenous and many patients are receiving immunosuppressive medications. Many mouse models exist, but these vary in the fidelity with which they model human disease. If the review is to advance our understanding, it should identify those findings that offer true mechanistic insights. In other words, where defects in the endocytic pathway are responsible for driving pathology rather than simple associations and correlations.
See for example, line 532:
Naïve CD4+ T cells can be skewed to become Tregs when their TCR is stimulated in the presence 532 of IL-2 and TGFβ219. The expansion of Tregs is greatly dependent on IL-2, which explains the 533 reduced number of Tregs in SLE patients220.
Treg abundance might be dependent on IL-2, and Tregs could be reduced in some patients with SLE, but this does not necessarily mean that IL-2 deficiency is the explanation.
Response: Citation #220 (citation #225 in the revision) (Malek, et al. 2002, Immunity) was earlier described as follows: “These observations indicate that the essential function of the IL-2/IL-2R system primarily lies at the level of the production of CD4(+)CD25(+) regulatory T cells.”
We have edited the above sentence as follows: “The expansion of Tregs is greatly dependent on IL-2, which suggests that IL-2 depletion contributes to the reduced numbers and function of Tregs in SLE patients”.
- Sections ostensibly concerned with outlining the basic mechanisms and function of endosomal recycling are conflated with aspects of disease pathogenesis. This might be unavoidable, but it means that the discussion of disease mechanisms is really far too brief for the reader to understand the mechanisms being discussed. See, for example, the telegraphic description of Sjogren’s syndrome, heart disease (lines 270-274).
Response: After careful consideration, we have decided to remove the description of other diseases in Section 2.1 since the title of this paper focuses on the pathogenesis of SLE. Instead, we added an example in SLE:
“T and B cell-derived exosomes have been found to be involved in SLE, for example, miR-451a expression in CD4+ T cell and B cell-derived exosomes is downregulated in SLE patients with active disease and it correlates with renal damage73.”
- The structure of the review is disorganised, which makes it hard to read or to understand. For example, Section 2.3 is ostensibly concerned with immune synapse formation. Indeed this section provides a brief overview of this complex topic. Section 3.1, which is under the topic of metabolic programming of T cells, discusses the immune synapse again, as well as the structure of the CD3 complex, and calcium flux. The possible relevance of this discussion is finally revealed at the end of section 3.1.1, where the authors discuss endocytic recycling of STIM1. This point should have been introduced much earlier to provide the reader with the context for the relevance of calcium flux.
Similarly, we have a long discussion about NAD hydrolases (Line 494). It is unclear why this discussion is included because at the end of the section, the authors report:
Since CD38 expression level on T cells impacts NAD+ levels, which is 494 used in a variety of redox and non-redox pathways, determining their functions199, determining 495 how its expression is controlled by endosomal recycling would be important. To date, what we 496 know thus far is that in human lymphocytes and T cell leukemia lines, the downregulation of CD38 497 expression by endocytosis is not a key step in its intracellular signaling, rather it is a negative 498 feedback mechanism200.
Response: We moved the discussions of immune synapse and normal TCR signaling from Section 3.1 to Section 2.3. Since calcium flux affects by both signaling through the TCR and STIM1 trafficking, we changed the title of section 3.1 to include both TCR and STIM1. In accordance with the reviewer’s suggestion, we eliminated the subsection 3.1.1.
We discussed the function of NAD hydrolases as NAD+ metabolism is critical for many aspects of T cell signaling. We have addressed the role of the specific NAD hydrolase CD38 because its surface expression is increased in SLE patients’ T cells and it is recruited to the immune synapse from recycling endosomes. The finding that endosome-bound CD38 does not mediate this metabolic signaling pathway suggests that recycling of CD38 back to the cell surface may contribute to this ectoenzyme function.
- It is surprising that so little attention is given to endosomal TLR receptors, ligations of which is known to be important in driving lupus pathogenesis.
Response: We expanded our review to address the role of endosome traffic of TLR2, TLR4, TLR-7, and TLR-8, which are expressed by CD4 T cells in a new section 3.5 entitled “Role of endosome traffic in TLR-mediated signaling”:
“Toll-like receptors (TLRs) are classified by the location of their expression; intracellular or extracellular. TLR2 and TLR4 are expressed on the surface of CD4+ T cells205 and when they bind to their ligand, bacterial membrane components206,207, they are internalized to the endosomes, a step required for the activation of nuclear factor-κB (NF-κB) and AP-1208,209. In endosomes, Rab11a promotes TLR4 signaling210 and Rab10 traffics TLR4 back to the cell surface211. Recently, it has been shown that Rab8 and Rab11 recruit adaptor protein-3 (AP-3) to TLR2 that promoted the secretion of IL-6 by phagocytic cells 212. TLR2 expression is also increased in SLE patients’ CD4+ T cells, CD8+ T cells, and B cells213.
TLR2 activation in human Tregs reduces their suppressive functions and it promotes Th17 differentiation in naïve CD4+ T cells 215. On the other hand, intracellular TLR7 and TLR8 confined to endosomes, where they sense viral single-stranded RNAs216. Genetic variants of TLR7 and TLR8 are risk factors for SLE 217,218, and their expression is upregulated in SLE patients219. Moreover, TLR7 activation in CD4+ T cells induces calcium flux220 and TLR8 signaling in human Tregs inhibits glucose uptake and glycolysis221.”
In section 4.2.1, we also address the role of TLR2 in IL-6 signaling:
“Majority of cells respond to IL-6 trans-signaling, which is through IL-6 binding to soluble IL-6 receptor (sIL-6R), rather than the classic signaling via IL-6R on the cell surface274. In human peripheral blood mononuclear cells, the cleavage of IL-6R to generate sIL-6R is induced by the activation of TLR2275.”
In section 4.2.3, we described how TLR2 upregulates the production of IL-17A and IL-17F:
“TLR2 activation in CD4+ T cells leads to pro-inflammatory skewing270. In-vitro stimulation of TLR2 on SLE patient CD4+ T cells with a synthetic bacterial lipopeptide, Pam3-Cys-Ser-Lys4 (Pam3CSK4), induces IL-17A and IL-17F production by upregulating H3K4 tri-methylation level in the IL-17A promoter region and H4 acetylation levels in both IL-17A and IL-17F promoter regions268.”
- There are countless errors of syntax. For example,
Line 248
“Rab35 is an regulates 248 rapid recycling of CD71 by localizing to the plasma membrane and early endosome58”.
Response: We have corrected it to read “Rab35 colocalizes with CD71 on the plasma membrane and vesicles, regulating the rapid recycling of CD71.”
- Several sections remain inexplicably highlighted.
Response: We have highlighted the revised sections of the manuscript for facilitating the review.
Reviewer 2 Report (New Reviewer)
The paper by Park JS et al. is a review, based on a large literature, concerning the link between endosomal traffic and CD4 lymphocyte activity in SLE. The text is easy to read and excellently implemented by figures and tables. However, I find the exclusion of CD8s from the paper somewhat limiting. Do the authors have an explanation for this?
Minor point:
Line 248: “Rab35 is an regulates rapid recycling of CD71….”, this sentence is not clear, please correct it.
Author Response
The paper by Park JS et al. is a review, based on a large literature, concerning the link between endosomal traffic and CD4 lymphocyte activity in SLE. The text is easy to read and excellently implemented by figures and tables. However, I find the exclusion of CD8s from the paper somewhat limiting. Do the authors have an explanation for this?
Minor point:
Line 248: “Rab35 is an regulates rapid recycling of CD71….”, this sentence is not clear, please correct it.
Response: We appreciate the reviewer’s positive feedback. While we recognize the importance of CD8+ T cells in SLE, we focused on CD4+ T cells since the expression of CD4 itself and of CD71 are regulated via endosome traffic. We have edited the syntax of line 248, which now reads “Rab35 colocalizes with CD71 on the plasma membrane and vesicles, regulating the rapid recycling of CD71.”
Reviewer 3 Report (New Reviewer)
General comments: The authors reviewed SLE pathogenesis from the cellular points of immunological view. The paper was fine and would provide a huge useful information to readers.
Specific comments:
1. Please discuss on the pathogenesis of SLE in response to belimumab and anifrolumab from the cellular points of immunological views like this review.
At least, you should discuss on the point in the manuscript.
2. IL-17 and Th17 cells are increased in the pathogenesis in not only acquired immunity but also innate immunity within plenty of autoimmune diseases.
How JAK/STAT3 or mTOR regulate Th17 cell differentiation? What is the trigger?
Author Response
General comments: The authors reviewed SLE pathogenesis from the cellular points of immunological view. The paper was fine and would provide a huge useful information to readers.
Specific comments:
- Please discuss on the pathogenesis of SLE in response to belimumab and anifrolumab from the cellular points of immunological views like this review.
Response: We appreciate the reviewer’s overall positive feedback. While belimumab and its target, BAFF, are important for immunotherapy in SLE, BAFF receptor is primarily expressed on B cells, and thus, we did not include it in our revision. However, the target of anifrolumab, type I IFN receptor, plays an important role in Tfh development, therefore, we have discussed therapeutic target at the end of Section 4.3.
“In-vitro, IL-6 and IL-21, which act through STAT3, drive Tfh differentiation334, however, in-vivo, IL-6 and IL-21 are not absolutely required for Tfh development335, suggesting the role of other cytokines in Tfh development. Type I IFN activates STAT1 to bind to B cell lymphoma 6 (Bcl6), a transcription factor that is required for Tfh development336, and in human and SLE mouse model, type I IFN signaling activates STAT4 to produce IL-21 and IFNγ337. In HeLa cells, Rab7 is required for early endosomal sorting of the subunit 1 of the type I IFN receptor (IFNAR1)338, and recently, anifrolumab, a monoclonal antibody targeting IFNAR1, has been FDA approved for SLE339. Therefore, endosome traffic of IFNAR1 in CD4+ T cells may modulate the clinical efficacy of this new therapeutic intervention in SLE.”
- At least, you should discuss on the point in the manuscript.
IL-17 and Th17 cells are increased in the pathogenesis in not only acquired immunity but also innate immunity within plenty of autoimmune diseases.
How JAK/STAT3 or mTOR regulate Th17 cell differentiation? What is the trigger?
Response: We discussed that the JAK/STAT3 pathway and mTOR promote Th17 differentiation in Sections 4.2.1 and 4.2.2, respectively. These pathways are also illustrated in Fig. 2.
This manuscript is a resubmission of an earlier submission. The following is a list of the peer review reports and author responses from that submission.
Round 1
Reviewer 1 Report
This manuscript is aiming at reviewing and discussing literature dealing with endosome trafficking and how it impacts T-cell signal transduction - which is really of interest. However, I do not feel that the content really matches with the initial aim, and specifically with the title provided: "endosome traffic modulates inflammatory signal transduction in CD4 T cells". The manuscript is focusing mainly (not to say exclusively) on systemic lupus erythematosus (SLE), and there is a vague and indirect link between endosome trafficking and CD4 T cell signal transduction. Indeed, few molecules in this manuscript are described to be differentially regulated via endocytic recycling leading to disturbed signal transduction and/or phenotype/differentiation. Most of the review is focusing on IL-2/IL-17 and Treg/Th17 differentiation in the context of SLE independently of endosome trafficking - which is misleading and confusing.
Aside, there is at least 12 occurrences citing reference 10 in the text, which is a paper of the authors’ manuscript published in 2009...even if this paper is a reference to argue that, in SLE, endosome trafficking might promote TCR zeta defects leading to the pathogenesis, it should not represent the foundation (together with one or two articles only) allowing to infer that T cell defects in SLE are the result of differential endosome trafficking. Maybe if the authors would not have focused only on the SLE field they would have found more papers in order to discuss more directly if and how “endosome traffic modulates inflammatory signal transduction in CD4 T cells”.
Author Response
Reviewer 1 comments:
This manuscript is aiming at reviewing and discussing literature dealing with endosome trafficking and how it impacts T-cell signal transduction - which is really of interest. However, I do not feel that the content really matches with the initial aim, and specifically with the title provided: "endosome traffic modulates inflammatory signal transduction in CD4 T cells". The manuscript is focusing mainly (not to say exclusively) on systemic lupus erythematosus (SLE), and there is a vague and indirect link between endosome trafficking and CD4 T cell signal transduction. Indeed, few molecules in this manuscript are described to be differentially regulated via endocytic recycling leading to disturbed signal transduction and/or phenotype/differentiation. Most of the review is focusing on IL-2/IL-17 and Treg/Th17 differentiation in the context of SLE independently of endosome trafficking - which is misleading and confusing.
Aside, there is at least 12 occurrences citing reference 10 in the text, which is a paper of the authors’ manuscript published in 2009...even if this paper is a reference to argue that, in SLE, endosome trafficking might promote TCR zeta defects leading to the pathogenesis, it should not represent the foundation (together with one or two articles only) allowing to infer that T cell defects in SLE are the result of differential endosome trafficking. Maybe if the authors would not have focused only on the SLE field they would have found more papers in order to discuss more directly if and how “endosome traffic modulates inflammatory signal transduction in CD4 T cells”.
Response: We appreciate that the reviewer pointed out the relative biases in our approach. In agreement with the suggestions, we have expanded our review to address the documented involvement of all Rab GTPases in endosome-mediated signal transduction in CD4 T cells beyond IL-2/IL-17 and Treg/Th17 differentiation. We extended out review to the involvement of Rab GTPases in STIM1 traffic to ORAI1 and calcium fluxing, production of exosomes, and role of endosome traffic in presentation of antigens to CD4 T cells. With respect to the impact of endosome traffic on the involvement of CD4 T cells in inflammation, this review is focused on SLE, as a natural model of pro-inflammatory signal transduction. SLE is a prototypical systemic autoimmune disorder that may serve as a model for inflammation for other autoimmune and infectious disease. To better reflect this approach, we include SLE in the title of the paper.
Reviewer 2 Report
The article is very usefull to understand the mechanism of SLE pathogesnesis, but the authors used a very scholar mood of presentation, even the subchpaters look as a book's chapter.
The big number of referrences is not so usefull taking in account that some data are from 1980/ 90.
Some phrases are unclear, with verbs.
so much abreviations please be asure that all of theme have been described extensivelly before.
the table must be reorganised with name of the study and year not with reference number
having the images in the end is very difficult to follow up, maybe on the top could be a better idea to put it
Maybe some medications must be described in order to see the utility of knowing the mechanism ( the ongoing studies- eventually)
Author Response
Reviewer 2 comments:
The article is very useful to understand the mechanism of SLE pathogesnesis, but the authors used a very scholar mood of presentation, even the subchpaters look as a book's chapter.
The big number of referrences is not so usefull taking in account that some data are from 1980/ 90.
Some phrases are unclear, with verbs.
so much abreviations please be asure that all of theme have been described extensivelly before.
the table must be reorganised with name of the study and year not with reference number
having the images in the end is very difficult to follow up, maybe on the top could be a better idea to put it
Maybe some medications must be described in order to see the utility of knowing the mechanism ( the ongoing studies- eventually)
Response: we appreciated the constructive comments, and, in response, we provided a i) comprehensive list of abbreviations on pages 3-4; ii) figures have been incorporated into a separate version of the manuscript labelled as manuscript with incorporated figures; and iii) we included medications with directly or indirectly affecting endosome traffic via inhibiting geranylgeranyl transferase II (3-PEHPC), mTOR (rapamycin, N acetylcysteine), ORAI1 (RO2959), or GLUT1 (CG-5).
Reviewer 3 Report
This article discusses the way endosome traffic modulates pro inflammatory signal transduction in CD4 T cells - very important and current aspect in the etiopathogenesis of many immune-mediated diseases. There are some comments I would like to make:
Lines 77-78 - "Cells that produce these pro-inflammatory cytokines cause autoimmune...." should be modified- autoimmunity does not necessarily develop, as it is suggested from this sentence, but under certain favorable conditions!
Also, in the introduction, the mention of pro- and anti-inflammatory cytokines should be accompanied by some examples
Line 79 - Il-2 is mentioned too abruptly and without a discussion about it
Rows 81-82 reference needed (FoxP3)
Tfh cells, HRES-1/Rab4, GTPase, ZAP-70, OXPHOS etc - explanation of the abbreviation
Row 184 “In SLE, T cell metabolism is dysregulated in multiple ways64” – please expand this affirmation especially in the conditions where in lines 99-99 you state "The following sections describe the disturbed balance of proinflammatory and anti-inflammatory cells in SLE and how they are regulated by their altered receptor endosomal recycling, resulting metabolic abnormalities, and downstream signaling.
Overall, the article, although well documented, seems to me very dense and difficult to follow. It would need more figures and schemes to be easier to understand. On the other hand, as long as the discussion focuses especially on the role of the various systems and cells in SLE, I think the title should be changed accordingly.
Author Response
Reviewer 3 comments:
This article discusses the way endosome traffic modulates pro inflammatory signal transduction in CD4 T cells - very important and current aspect in the etiopathogenesis of many immune-mediated diseases. There are some comments I would like to make:
Response: we appreciated the overall positive comments. Each specific comment is separately addressed below:
Comment: Lines 77-78 - "Cells that produce these pro-inflammatory cytokines cause autoimmune...." should be modified- autoimmunity does not necessarily develop, as it is suggested from this sentence, but under certain favorable conditions!
Response: the referenced sentence was modified as follows: “Cells that produce these pro-inflammatory cytokines contribute to autoimmune diseases, including rheumatoid arthritis and systemic lupus erythematosus (SLE)14,15.”
Comment: Also, in the introduction, the mention of pro- and anti-inflammatory cytokines should be accompanied by some examples
Line 79 - Il-2 is mentioned too abruptly and without a discussion about it
Response: To create a transition, we revised the introduction of IL-2 as follows: “In contrast to proinflammatory cytokines, interleukin 2 (IL-2) plays a unique role as an anti-inflammatory cytokine given its fundamental role in regulatory T cell (Treg) differentiation16.”
Comment: Rows 81-82 reference needed (FoxP3).
Response: The following reference has been added on the role of FoxP3 in development and function of Tregs19.
Comment: Tfh cells, HRES-1/Rab4, GTPase, ZAP-70, OXPHOS etc - explanation of the abbreviation.
Response: A list of abbreviation has been provided on pages 3-4.
Comment: Row 184 “In SLE, T cell metabolism is dysregulated in multiple ways64” – please expand this affirmation especially in the conditions where in lines 99-99 you state "The following sections describe the disturbed balance of proinflammatory and anti-inflammatory cells in SLE and how they are regulated by their altered receptor endosomal recycling, resulting metabolic abnormalities, and downstream signaling.
Response: Rows 288-483 have been dedicated to address the influence of endosome traffic on metabolism, involving metabolic enzymes, such as CD38 and transporters, such as CD71 and CD98.
Comment: Overall, the article, although well documented, seems to me very dense and difficult to follow. It would need more figures and schemes to be easier to understand. On the other hand, as long as the discussion focuses especially on the role of the various systems and cells in SLE, I think the title should be changed accordingly.
Response: the title of the paper was revised as follows, “ENDOSOME TRAFFIC MODULATES PRO-INFLAMMATORY SIGNAL TRANSDUCTION IN CD4 T CELLS – IMPLICATIONS FOR THE PATHOGENESIS OF SYSTMEMIC LUPUS ERYTHEMATOSUS”
Round 2
Reviewer 1 Report
Authors made some significant improvements. But, in order to support/match with the title, the manuscript still lacks references showing a direct evidence of differential endosomal trafficking leading to differential T-cell signaling: except for CD71 and CD3zeta, the rest is mostly indirect.